# Spectral Imagery Tensor Decomposition for Semantic Segmentation of Remote Sensing Data through Fully Convolutional Networks

**Josué López [1,*]**, **Deni Torres [1]**, **Stewart Santos [2]** and **Clement Atzberger [3]**

1    Center for Research and Advanced Studies of the National Polytechnic Institute, Telecommunications
     Group, Av del Bosque 1145, Zapopan 45017, Mexico; dtorres@gdl.cinvestav.mx
2    University of Guadalajara, Center of Exact Sciences and Engineering, Blvd. Gral. Marcelino García
     Barragán 1421, Guadalajara 44430, Mexico; stewart.santos@academicos.udg.mx
3    University of Natural Resources and Life Science, Institute of Geomatics, Peter Jordan 82,
     Vienna 1180, Austria; clement.atzberger@boku.ac.at
*    Correspondence: jalopez@gdl.cinvestav.mx

**Abstract:** This work aims at addressing two issues simultaneously: data compression at input space and semantic segmentation. Semantic segmentation of remotely sensed multi- or hyperspectral images through deep learning (DL) artificial neural networks (ANN) delivers as output the corresponding matrix of pixels classified elementwise, achieving competitive performance metrics. With technological progress, current remote sensing (RS) sensors have more spectral bands and higher spatial resolution than before, which means a greater number of pixels in the same area. Nevertheless, the more spectral bands and the greater number of pixels, the higher the computational complexity and the longer the processing times. Therefore, without dimensionality reduction, the classification task is challenging, particularly if large areas have to be processed. To solve this problem, our approach maps an RS-image or third-order tensor into a core tensor, representative of our input image, with the same spatial domain but with a lower number of new tensor bands using a Tucker decomposition (TKD). Then, a new input space with reduced dimensionality is built. To find the core tensor, the higher-order orthogonal iteration (HOOI) algorithm is used. A fully convolutional network (FCN) is employed afterwards to classify at the pixel domain, each core tensor. The whole framework, called here HOOI-FCN, achieves high performance metrics competitive with some RS-multispectral images (MSI) semantic segmentation state-of-the-art methods, while significantly reducing computational complexity, and thereby, processing time. We used a Sentinel-2 image data set from Central Europe as a case study, for which our framework outperformed other methods (included the FCN itself) with average pixel accuracy (PA) of 90% (computational time $\sim$90s) and nine spectral bands, achieving a higher average PA of 91.97% (computational time $\sim$36.5s), and average PA of 91.56% (computational time $\sim$9.5s) for seven and five new tensor bands, respectively.

**Keywords:** fully convolutional network; semantic segmentation; spectral image; tensor decomposition

## 1. Introduction

Remote sensing RS images are of great use in many earth observation applications, such as agriculture, forest monitoring, disaster prevention, security affairs, and others [1]. The recent and upcoming availability of multispectral and hyperspectral satellites alleviates specific tasks, such as detection, classification, and semantic segmentation. In semantic segmentation, also called pixel-wise classification, each pixel in an RS image is assigned to one class [1]. This classification becomes easier when higher dimensional spectral information is acquired [1]. Spectral systems split, by physical filters,

the incoming radiance, and provide a vector with spectral reflectance values called spectral signatures. The remotely sensed spectral signatures enable a precise interpretation and recognition of different elements of interest covering the earth surface [2].

Supervised and unsupervised classification of RS images is a very active research area in spectral analysis [3]. To reduce the data dimensionality, and to concentrate the information into a fewer number of features, a once widely used approach was to define various indices to facilitate the classification of diverse land cover [4]. For instance, normalized difference vegetation index (NDVI) [5] and normalized difference water index (NDWI) [6] use a combination of visible to near infrared (NIR) spectral reflectance respectively, to assess land cover, vegetation vitality, and water status [4]. Additionally, supervised machine learning techniques such as random forest [7], support vector machine (SVM) [8,9], decision trees [10], and ANN [11] have been used for RS spectral image classification and have achieved very high accuracy rates [12]. More recently, CNN has been used for semantic segmentation of multispectral images (MSI), promising to be an alternative for solving semantic segmentation issues [13].

The high spectral redundancy of spectral images produces a huge unnecessary number of computations in classification/segmentation algorithms. It is therefore advisable to implement these algorithms together with a dimensionality reduction preprocessing [14]. Spectral data are stored as three-dimensional arrays, so it seems possible to use tensor decomposition (TD) methods [15] for preprocessing, to reduce high redundancy while avoiding information loss [14]. Different to matrix-based decomposition algorithms, such as principal components analysis (PCA) [16] and SVD, TD approach allows to treat spectral data as third-order tensor preserving the spatial information, which sustains the pixel-wise classification task.

In this work we aim addressing two main issues: data compression at input space, and semantic segmentation; i.e., pixel-wise classification of RS imagery. We introduce a spectral data preprocessing that preserves tensor structure and reduces information loss through tensor algebra [17], with the ultimate aim of reducing processing time while keeping high accuracy in further semantic segmentation CNNs. This will produce MSI compression, preserving the spatial domain while reducing the spectral domain, decomposing the original tensor into a core tensor with same order but much lower dimensionality multiplied by a matrix in each mode in the context of tensor algebra [17]. The core tensor, with lower rank than the original data, is used as the input data to the semantic segmentation ANN instead of the MSIs, decreasing the number of computations and in turn the execution time. Previous experimental results demonstrate high performance in semantic segmentation with circa $10\times$ speed up in execution time [18].

The proposed framework can be applied to multispectral, hyperspectral, and even multitemporal datasets. As a particular case, in this study we performed experiments using RS multispectral dataset from the european space agency (ESA) program Sentinel-2 [19] with five classes (soil, water, vegetation, cloud, and shadow).

### 1.1. Related Work

In recent years, spectral data for earth surface classification has been a very active research area. Methods proposed by Kemker et al. [11,20], Hamida et al. [21], and López et al. [18] use CNNs for RS-CNNMSI pixel-wise classification. Nevertheless, processing raw spectral data with deep learning (DL) algorithms is computationally very expensive. Wang et al. [22] introduced a salient band selection method for HSIs by manifold ranking, and Li et al. [23] proposed a band selection method from the perspective of spectral shape similarity analysis of RS-HSIs to obtain less computational complexity. However, some surface materials differentiate from each other in specific bands, so cutting off spectral bands negatively affected further classification tasks.

More recently, the use of tensor approach for spectral images compression has been introduced; see Zhang et al. [24]. Many authors adopted dimensionality reduction algorithms, such as PCA [16] and singular value decomposition (SVD), for spectral image compression. Other authors have made efforts

to reduce the computational cost in CNNs for image classification by using TD algorithms [25,26]. Astrid et al. in [25] proposed a CNN compression method based on CPD and the tensor power method where they achieved significant reduction in memory and computational cost. Chien et al. in [26] presents a tensor-factorized ANN, which integrates TD and ANNs for multi-way feature extraction and classification. Nevertheless, although the idea is to compress data in order to reduce computational cost and processing time, these works compress or decompose the data of the hyper-parameters within the network, which causes the training of the semantic segmentation or classification network to be slower due to the change of the weights in the tensor decomposition.

Recently, three works close to our research [27–29] were published. In [27] An et al. proposed an unsupervised tensor-based multiscale low rank decomposition (T-MLRD) method for hyperspectral image dimensionality reduction, and Li et al. in [28] proposed a low-complexity compression approach for multispectral images based on convolution neural networks CNNs with nonnegative Tucker decomposition (NTD). Nevertheless, these methods reduce the tensor in every dimension, which is self-defeating for a segmentation CNN. Besides, the non-negative decomposed tensor proposed in [28] causes slower convergence in DL algorithms. In [29] An et al. proposed a tensor discriminant analysis (TDA) model via compact feature representation, wherein the traditional linear discriminant analysis was extended to tensor space to make the resulting feature representation more discriminant. However, this approach still leads to a degradation of the spatial resolution, which disturbed the CNN performance. See Table 1 for a summary of the related works.

**Table 1.** Related work in spectral imagery semantic segmentation.

| Reference | Input | Decomposition | Reduction | Classifier |
|---|---|---|---|---|
| Li, S. et al. [23] (2014) | HSI | - | Band selection | SVM |
| Zhang, L. et al. [24] (2015) | HSI | TKD | Spatial-Spectral | - |
| Wan, Q. et al. [22] (2016) | HSI | - | Band selection | SVM/kNN/CART |
| Kemke, R. et al. [11] (2017) | MSI | - | - | CNN |
| Hamida, A. et al. [21] (2017) | MSI | - | - | CNN |
| Li, J. et al. [28] (2019) | MSI | NTD-CNN | Spatial-spectral | - |
| An, J. et al. [27] (2019) | HSI | T-MLRD | Spatial-spectral | SVM/1NN |
| An, J. et al. [29] (2019) | HSI | TDA | Spatial-spectral | SVM/1NN |
| Our framework (2019) | MSI | HOOI | Spectral | FCN |

*1.2. Contribution*

The contribution of this work is summarized into three main points:

1. RS-CNNMSI or -HSI, or third order tensors are compressed in the spectral domain through TKD preprocessing, preserving the pixel spatial structure and obtaining a core tensor representative of the original. These core tensors, with less new tensor bands, which belong to subspaces of the original space, build the new input space for any supervised classifier at pixel level, which delivers the corresponding prediction matrix of pixels classified element-wise. This approach achieves high or competitive performance metrics but with less computational complexity, and consequently, lower computational time.

2. This approach outperforms other methods in normalized difference indexes, PCA, particularly the same FCN with original data. Each core tensor is calculated using the HOOI algorithm, which achieves high orthogonality degree for the core tensor (all-orthogonality) and for its factor matrices (column-wise orthogonal); besides, it converges faster than others, such as TUCKALS3 [17].

3. The efficiency of this approach can be measured by one or more performance metrics, e.g., pixel accuracy (PA), as a function of the number of new tensor bands, orthogonality degree of the factor matrices and the core tensor, reconstruction error of the original tensor, and execution time. These results are shown in Section 6: Experimental Results.

The remainder of this work is organized as follows. Section 2 introduces tensor algebra notation and basic concepts to familiarize the reader with the symbology used in this paper. Section 3 presents the problem statement of this work and the mathematical definition. In Section 4, CNN theory is described for classification and semantic segmentation. Section 5 presents the framework proposed for compression and semantic segmentation of spectral images. Experimental results are presented in Section 6. Finally, Sections 7 and 8 present a discussion and conclusions based on the results obtained in the experiments.

## 2. Tensor Algebra Basic Concepts

For this work we used the conventional tensor algebra notation [15]. Hence, scalars or zero order tensors are represented by italic lowercase letters; e.g., $a$. Vectors or first order tensor are denoted by boldface lowercase letters; e.g., $\mathbf{a}$. Matrices or tensor of order two are denoted by boldface capital letters, e.g., $\mathbf{A}$, and three or higher order tensors by boldface Euler script letters, e.g., $\mathcal{A}$. In a $N$-order tensor $\mathcal{A} \in \mathbb{R}^{I_1 \times \cdots \times I_N}$, where $\mathbb{R}$ represents the set of real numbers, $I_n$ indicates the size of the tensor in each mode $n = \{1, \ldots, N\}$. An element of $\mathcal{A}$ is denoted with indices in lowercase letters, e.g., $a_{i_1 \ldots i_N}$ where $i_n$ denotes the $n$-mode of $\mathcal{A}$ [17]. A fiber is a vector, the result of fixing every index of a tensor but one, and it is denoted by $\mathbf{a}_{:i_2 i_3}$, $\mathbf{a}_{i_1 : i_3}$, and $\mathbf{a}_{i_1 i_2}$—for column, row, and tube fibers respectively for a third order tensor instance. A slice is a matrix, the result of fixing every index of a tensor but two, and it is denoted by $\mathbf{A}_{i_1::}$, $\mathbf{A}_{:i_2:}$, and $\mathbf{A}_{::i_3}$, or more compactly, $\mathbf{A}_{i_1}$, $\mathbf{A}_{i_2}$, and $\mathbf{A}_{i_3}$ for horizontal, lateral, and frontal slices respectively for a third order tensor instance. Finally, $\mathbf{A}^{(n)}$ denotes a matrix element from a sequence of matrices [17].

It is also necessary to introduce some tensor algebra operations and basic concepts used in later explanations. These notations were taken textually from [17].

### 2.1. Matricization

The mode-$n$ matricization is the process of reordering the elements of a tensor into a matrix along axis $n$ and it is denoted as $\mathbf{A}_{(n)} \in \mathbb{R}^{I_n \times \prod_{m \neq n} I_m}$.

### 2.2. Outer Product

The outer product of $N$ vectors $\mathcal{X} = \mathbf{a}^{(1)} \circ \cdots \circ \mathbf{a}^{(N)}$ produces a tensor $\mathcal{X} \in \mathbb{R}^{I_1 \times \cdots \times I_N}$ where $\circ$ denotes the outer product and $\mathbf{a}^{(n)}$ denotes a vector in a sequence of $N$ vectors and each element of the tensor is the product of the corresponding vector elements; i.e., $x_{i_1 i_2 \ldots i_N} = a_{i_1}^{(1)} \ldots a_{i_N}^{(N)}$.

### 2.3. Inner Product

The inner product of two tensors $\mathcal{A}, \mathcal{B} \in \mathbb{R}^{I_1 \times \cdots \times I_N}$ is the sum of the products of their entries; i.e., $\langle \mathcal{A}, \mathcal{B} \rangle = \sum_{i_1=1}^{I_1} \cdots \sum_{i_N=1}^{I_N} a_{i_1 \ldots i_N} b_{i_1 \ldots i_N}$.

### 2.4. N-Mode Product

It means the multiplication of a tensor $\mathcal{A} \in \mathbb{R}^{I_1 \times \cdots \times I_N}$ by a matrix $\mathbf{U} \in \mathbb{R}^{J \times I_n}$ or vector $\mathbf{u} \in \mathbb{R}^{I_n}$ in mode $n$; i.e., along axis $n$. It is represented by $\mathcal{B} = \mathcal{A} \times_n \mathbf{U}$, where $\mathcal{B} \in \mathbb{R}^{I_1 \times \cdots \times I_{n-1} \times J \times I_{n+1} \times \cdots \times I_N}$ [17].

### 2.5. Rank-One Tensor

A tensor $\mathcal{X} \in \mathbb{R}^{I_1 \times \cdots \times I_N}$ is rank one if it can be written as the outer product of $N$ vectors; i.e., $\mathcal{X} = \mathbf{a}^{(1)} \circ \cdots \circ \mathbf{a}^{(N)}$.

*2.6. Rank-R Tensor*

The rank of a tensor rank($\mathcal{X}$) is the smallest number of components in a CPD; i.e., the smallest number of rank-one tensors that generate $\mathcal{X}$ as their sum [17].

*2.7. N-Rank*

The *n*-rank of a tensor $\mathcal{X} \in \mathbb{R}^{I_1 \times \cdots \times I_N}$ denoted $\text{rank}_n(\mathcal{X})$, is the column rank of $\mathbf{X}_{(n)}$; i.e., the dimension of the vector space spanned by the mode-*n* fibers. Hence, if $R_n \equiv \text{rank}_n(\mathcal{X})$ for $n = 1, \ldots, N$, we can say that $\mathcal{X}$ has a rank $-(R_1, \ldots, R_N)$ tensor.

All the tensor algebra notation presented until this point is summarized in Table 2 for simpler regarding.

**Table 2.** Tensor algebra notation summary

| | |
|---|---|
| $\mathcal{A}, \mathbf{A}, \mathbf{a}, a$ | Tensor, matrix, vector and scalar respectively |
| $\mathcal{A} \in \mathbb{R}^{I_1 \times \cdots \times I_N}$ | *N*-order tensor of size $I_1 \times \cdots \times I_N$. |
| $a_{i_1 \ldots i_N}$ | An element of a tensor |
| $\mathbf{a}_{:i_2 i_3}, \mathbf{a}_{i_1 : i_3}$, and $\mathbf{a}_{i_1 i_2 :}$ | Column, row and tube fibers of a third order tensor |
| $\mathbf{A}_{i_1 ::}, \mathbf{A}_{:i_2:}, \mathbf{A}_{::i_3}$ | Horizontal, lateral and frontal slices for a third order tensor |
| $\mathbf{A}^{(n)}, \mathbf{a}^{(n)}$ | A matrix/vector element from a sequence of matrices/vectors |
| $\mathbf{A}_{(n)}$ | Mode-n matricization of a tensor. $\mathbf{A}_{(n)} \in \mathbb{R}^{I_n \times \prod_{m \neq n} I_m}$ |
| $\mathcal{X} = \mathbf{a}^{(1)} \circ \cdots \circ \mathbf{a}^{(N)}$ | Outer product of *N* vectors, where $x_{i_1 i_2 \ldots i_N} = a_{i_1}^{(1)} \ldots a_{i_N}^{(N)}$ |
| $\langle \mathcal{A}, \mathcal{B} \rangle$ | Inner product of two tensors. |
| $\mathcal{B} = \mathcal{A} \times_n \mathbf{U}$ | *n*-mode product of tensor $\mathcal{A} \in \mathbb{R}^{I_1 \times \cdots \times I_N}$ by a matrix $\mathbf{U} \in \mathbb{R}^{J \times I_n}$ along axis *n*. |

*2.8. Tucker Decomposition (Tkd)*

The TKD can be seen as a form of higher-order PCA [17]. This method decomposes a tensor $\mathcal{X} \in \mathbb{R}^{I_1 \times \cdots \times I_N}$ into a core tensor $\mathcal{G} \in \mathbb{R}^{J_1 \times \cdots \times J_N}$ multiplied by a matrix along each mode $n = 1, \ldots, N$ as

$$\mathcal{X} \approx \mathcal{G} \times_1 \mathbf{U}^{(1)} \cdots \times_N \mathbf{U}^{(N)} \tag{1}$$

where the core tensor preserves the level of interaction for each factor or projection matrix $\mathbf{U}^{(n)} \in \mathbb{R}^{I_n \times J_n}$. These matrices are usually, but not necessarily, orthogonal, and can be thought of as the principal components in each mode [17] (see Figure 1). $J_n$ represents the number of components in the decomposition; i.e., the rank $-(R_1, \ldots, R_N)$. We compute rank $-(R_1, \ldots, R_N)$, where $\text{rank}_n(\mathcal{X}) = R_n$ for every *n*-mode, which generally does not exactly reproduce $\mathcal{X}$. Starting from (1), the reconstruction of an approximated tensor can be given by where $\hat{\mathcal{X}}$ is the reconstructed tensor. Then, we can acquire the core tensor $\mathcal{G}$ by the multilinear projection

$$\mathcal{G} = \mathcal{X} \times_1 \mathbf{U}^{(1)\text{T}} \cdots \times_N \mathbf{U}^{(N)\text{T}}, \tag{2}$$

where $\mathbf{U}^{(n)\text{T}}$ denotes the transpose matrix of $\mathbf{U}^{(n)}$ for $n = 1, \ldots, N$. The reconstruction error $\xi$ can be computed as

$$\xi(\hat{\mathcal{X}}) = ||\mathcal{X} - \hat{\mathcal{X}}||_F^2, \tag{3}$$

where $||\cdot||_F$ represents the Frobenius norm. To effectively compress data, the reconstructed lower-rank tensor $\hat{\mathcal{X}}$ should be close to the original tensor $\mathcal{X}$; this can be reached by an algorithm as HOOI, which is iterative, and it is described in Section 5.1.

$$\hat{\mathcal{X}} = \mathcal{G} \times_1 \mathbf{U}^{(1)} \cdots \times_N \mathbf{U}^{(N)}, \tag{4}$$

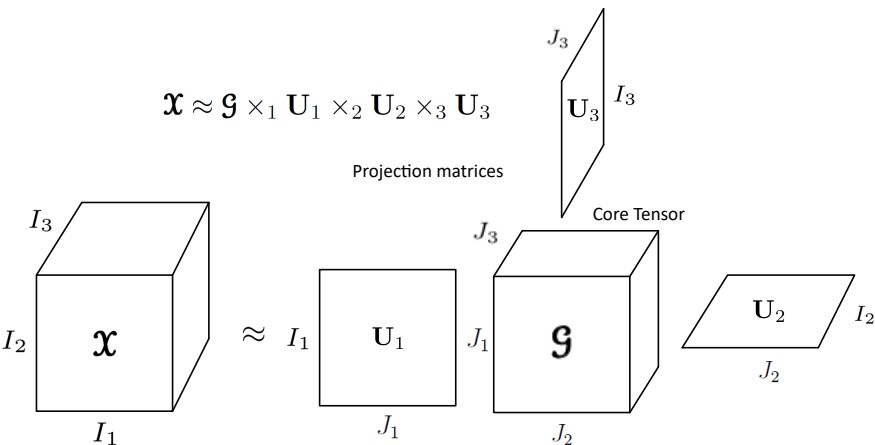

**Figure 1.** Tucker decomposition for a third-order tensor.

## 3. Problem Statement and Mathematical Definition

Spectral images are third-order arrays, which provide not only spatial, but also spectral features from RS scenes of interest. These properties aid CNNs to easily find features to characterize the behaviors of different materials over the earth's surface. However, the large amount of spectral data causes huge computational load, and therefore, large processing time using machine learning algorithms.

It is important to preserve the three-dimensional array structure of the RS spectral input image, in order to effectively classify each pixel of the image. In RS multi- or hyperspectral images, the spectral bands are highly correlated, and contain lot of redundancy. Therefore, we propose a TKD-based method as a preprocessing step to provide a better suited input for the semantic segmentation based on CNN. This will also considerably reduce high number of parameters, and in turn, processing time during training and testing. Our problem statement for RS spectral images can be described as follows.

### 3.1. Problem Statement

Given a pair $(\mathcal{X}, \mathbf{Y})$, where tensor $\mathcal{X} \in \mathbb{R}^{I_1 \times I_2 \times I_3}$ denotes a CNNMSI or HSI, and $\mathbf{Y} \in \mathbb{R}^{I_1 \times I_2}$ its corresponding ground truth matrix for a specific number of classes $C$, find another pair $(\mathcal{G}, \hat{\mathbf{Y}})$, where the tensor $\mathcal{G} \in \mathbb{R}^{J_1 \times J_2 \times J_3}$, used for classification, is representative of $\mathcal{X}$, and $\hat{\mathbf{Y}}$ is its associated matrix of predicted classes; preserving the spatial-domain $J_1 = I_1$, $J_2 = I_2$ but with fewer new tensor bands, i.e., $J_3 < I_3$, achieving higher or competitive performance metrics for pixel-wise classification, reducing the dimensionality, and therefore, decreasing computational complexity in the classification task.

### 3.2. Mathematical Definition

We can describe the problem stated in previous subsection mathematically as the following optimization problem

$$
\begin{aligned}
\min_{\mathcal{G}, \mathbf{U}^{(1)}, \mathbf{U}^{(2)}, \mathbf{U}^{(3)}} \quad & ||\mathcal{X} - \mathcal{G} \times_1 \mathbf{U}^{(1)} \times_2 \mathbf{U}^{(2)} \times_3 \mathbf{U}^{(3)}||_F^2 \\
\text{subject to} \quad & \mathbf{U}^{(n)} \in St_{I_n \times J_n} \quad \text{and} \quad St_{I_n \times J_n} \equiv \{\mathbf{U}^{(n)} \in \mathbb{R}^{I_n \times J_n} \mid \mathbf{U}^{(n)\mathrm{T}}\mathbf{U}^{(n)} = \mathbf{I}^{(n)}\}, \\
& J_1 = I_1, J_2 = I_2 \qquad \text{preserving the pixel domain,} \\
& J_3 < I_3 \qquad\qquad \text{reducing spectral dimensionality} \\
& \xi(\hat{\mathcal{X}}) \leq \psi \qquad\qquad \text{mesaure of how representative of } \hat{\mathcal{X}} \ \mathcal{G} \text{ is}
\end{aligned}
\tag{5}
$$

where $\psi$ denotes an error threshold defined depending on the accuracy or performance metrics required for each application and $St_{I_n \times J_n}$ represents the Stiefel manifold [30]. Embedding $\mathcal{G}$ into the

objective function, as Lathhauwer proved in [31] Theorems 3.1, 4.1, and 4.2, (5), can be written by the equivalent under the same constraints as (5).

$$\max_{\mathbf{U}^{(1)},\mathbf{U}^{(2)},\mathbf{U}^{(3)}} ||\boldsymbol{\mathcal{X}} \times_1 \mathbf{U}^{(1)\mathrm{T}} \times_2 \mathbf{U}^{(2)\mathrm{T}} \times_3 \mathbf{U}^{(3)\mathrm{T}}||_F^2 \tag{6a}$$

$$\text{where} \quad \boldsymbol{\mathcal{G}} = \boldsymbol{\mathcal{X}} \times_1 \mathbf{U}^{(1)\mathrm{T}} \times_2 \mathbf{U}^{(2)\mathrm{T}} \times_3 \mathbf{U}^{(3)\mathrm{T}} \tag{6b}$$

The subtensors $\boldsymbol{\mathcal{G}}_{i_n}$ of the core tensor $\boldsymbol{\mathcal{G}}$ satisfy the all-orthogonality property [32], which establishes that two subtensors $\boldsymbol{\mathcal{G}}_{i_n=\alpha}$ and $\boldsymbol{\mathcal{G}}_{i_n=\beta}$ are all-orthogonal

$$\langle \boldsymbol{\mathcal{G}}_{i_n=\alpha}, \boldsymbol{\mathcal{G}}_{i_n=\beta} \rangle = 0 \tag{7}$$

for all possible values of $n$, $\alpha$, and $\beta$ subject to $\alpha \neq \beta$, and the ordering property:

$$\|\boldsymbol{\mathcal{G}}_{i_n=1}\|_F \geq \|\boldsymbol{\mathcal{G}}_{i_n=2}\|_F \geq \cdots \geq \|\boldsymbol{\mathcal{G}}_{i_n=I_N}\|_F. \tag{8}$$

Our optimization problem can be solved by several algorithms. In this work, the HOOI algorithm was selected (described in Section 5.1), due to its convergence and orthogonality performance. Once a tensor $\boldsymbol{\mathcal{G}}$ is obtained, a classifier $f$ that belongs to the hypothesis space $H$ maps input data $\boldsymbol{\mathcal{G}}$ into output data $\hat{\mathbf{Y}}$; that is

$$\hat{\mathbf{Y}} = f(\boldsymbol{\mathcal{G}}) \tag{9}$$

where $f$ is a pixel-wise classifier. In this paper, a FCN for semantic segmentation was used as classifier due to the need of classify each pixel of the input image and to its performance in pixel accuracy. The FCN used in this work is described in Section 4.

## 4. Convolutional Neural Networks (CNNs)

CNNs are supervised feed-forward DL-ANNs for computer vision. The idea of applying a sort of convolution of the synaptic weights of a neural network through the input data yields to a preservation of spatial features, which alleviates the hard task of classification and in turn semantic segmentation. This type of ANN works under the same linear regression model as every machine learning (ML) algorithm. Since images are three dimensional arrays, we can use tensor algebra notation to describe the input of CNNs as a tensor $\boldsymbol{\mathcal{A}} \in \mathbb{R}^{I_1 \times I_2 \times I_3}$, where $I_1$, $I_2$, and $I_3$ represent height, width, and depth of the third order array respectively; i.e., the spatial and spectral domain of an image. We can write generally the linear regression model used for ANNs as

$$\hat{\mathbf{y}} = \sigma\left(\mathbf{W}\mathbf{g} + \mathbf{b}\right) \tag{10}$$

where $\hat{\mathbf{y}}$ represents the output prediction of the network; $\sigma$ denotes an activation function; $\mathbf{g}$ is the input dataset; $\mathbf{W}$ and $\mathbf{b}$ are the matrix of synaptic weights and the bias vector, respectively. These parameters are adjustable; i.e., their values are modified every iteration looking for convergence to minimize the loss in the prediction through optimization algorithms [33]. For simplicity, the bias vector can be ignored, assuming that matrix $\mathbf{W}$ will update until convergence independently of another parameter [33]. Considering that the input dataset to a CNN is a multidimensional array, we can represent (9) and (10) using tensor algebra notation as

$$\hat{\boldsymbol{y}} = \sigma\left(\boldsymbol{\mathcal{W}}\boldsymbol{\mathcal{G}}\right) \tag{11}$$

where $\hat{\boldsymbol{y}}$ represents the prediction output tensor of the ANN (in our case, a second order tensor or matrix $\hat{\mathbf{Y}}$), $\boldsymbol{\mathcal{G}}$ is the input dataset, and $\boldsymbol{\mathcal{W}}$ is a $K_1 \times K_2 \times F_1$ tensor called filter or kernel with the adaptable synaptic weights. Different to conventional ANN, in CNNs, $\boldsymbol{\mathcal{W}}$ is a shiftable square tensor is much

smaller in height and width than the input data, i.e., $K_1 = K_2$ and $K_s << I_s$ for $s = 1, 2$; $F_1$ denotes the number of input channels; i.e., $F_1 = I_3$. For hidden layers, instead of the prediction tensor $\hat{\mathcal{y}}$, the output is a matrix called activation map $\mathbf{M} \in \mathbb{R}^{I_1 \times I_2}$, which preserves features from the original data in each domain. Actually, it is necessary to use much kernels $\mathcal{W}^{(f_2)}$ as activation maps, with different initialization values to preserve diverse features of the image. Hence, we can also define activation maps as a tensor $\mathcal{M} \in \mathbb{R}^{I_1 \times I_2 \times F_2}$ where $F_2$ denotes the number of activation maps produced by each filter (see Figure 2). Kernels are displaced through the whole input image as a discrete convolution operation. Then, each element of the output activation map $m_{i_1 i_2 f_2}$ is computed by the summary of the Hadamard product of kernel $\mathcal{W}^{(f_2)}$ and a subtensor from the input tensor $\mathcal{G}$ centered in position $(i, j)$ and with same dimensions of $\mathcal{W}$, as follows

$$m_{i_1 i_2 f_2} = \sigma \left[ \sum_{k_1=1}^{K_1} \sum_{k_2=1}^{K_2} \sum_{f_1=1}^{F_1} w_{k_1, k_2, f_1} g_{i_1+k_1-o_1, i_2+k_2-o_2, f_1} \right] \tag{12}$$

where $m_{i_1 i_2 f_2}$ denotes the value of the output activation map $f_2$ at position $i_1, i_2$; $\sigma$ represents the activation function; and $o_1$ and $o_2$ are offsets in spatial dimensions which depend on the kernel size, and equal $\frac{K_1+1}{2}$ and $\frac{K_2+1}{2}$ respectively (see Figure 2).

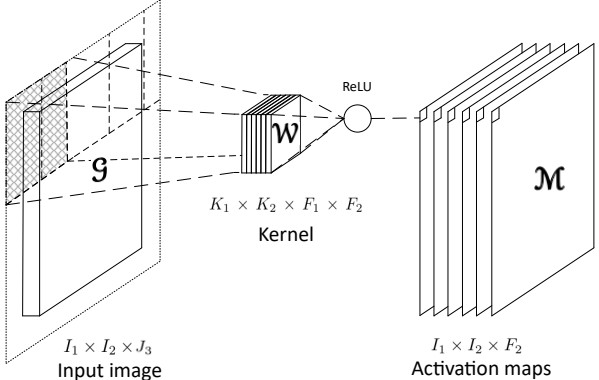

**Figure 2.** Convolutional layer with a $K_1 \times K_2 \times F_1 \times F_2$ kernel. Input channels $F_1$ must equal the spectral bands $I_3$. To preserve original dimensions at the output, zero padding is needed [18]. Output dimensions also depend on stride $S = 1$ to consider every piece of pixel information and to preserve original dimensions.

An ANN is trained by using iterative gradient-based optimizers, such as Stochastic gradient descent, Momentum, RMSprop, and Adam [33]. This drive the cost function $L(\mathcal{W})$ to a very low value by updating the synaptic weights $\mathcal{W}$. We can compute the cost function by any function that measures the difference between the training data and the prediction, such as Euclidean distance or cross-entropy [10]. Besides, the same function is used to measure the performance of the model during testing and validation. In order to avoid overfitting [33], the total cost function used to train an ANN combines one of the cost functions mentioned before, plus a regularization term.

$$J(\mathcal{W}) = L(\mathcal{W}) + R(\mathcal{W}), \tag{13}$$

where $J(\mathcal{W})$ denotes the total cost function and $R(\mathcal{W})$ represents a regularization function. Then, we can decrease $J(\mathcal{W})$ by updating the synaptic weights in the direction of the negative gradient. This is known as the method of steepest descent or gradient descent.

$$\mathcal{W}' = \mathcal{W} - \alpha \nabla_{\mathcal{W}} J(\mathcal{W}), \tag{14}$$

where $\mathcal{W}'$ represents the synaptic weights tensor in next iteration during training, $\alpha$ denotes the learning rate parameter, and $\nabla_{\mathcal{W}} J(\mathcal{W})$ the cost function gradient. Gradient descent converges when every element of the gradient is zero, or in practice, very close to zero [10].

CNNs has been successfully used in many image classification frameworks. This variation in architecture from other typical ANN models yields the network to learn spatial and spectral features, which are highly profitable for image classification. Besides, FCNs, constructed with only convolutional layers are able to classify each element of the input image; i.e., they yield pixel-wise classification, or in other words, semantic segmentation.

## 5. Hooi-Fcn Framework

In this work we propose a TKD-CNN-based framework called HOOI-FCN, which maps the original high-correlated spectral image into a low-rank core tensor, preserving enough statistical information to alleviate image pixel-wise classification. The aim is to improve performance while reducing processing time in semantic segmentation ANNs by compressing CNNMSI third-order tensors. Applying TD methods, relevant information is preserved, mainly acquired from the spectral domain, convenient for the classification FCN. This novel framework is in summary, a two step structure composed by an HOOI TD and a FCN for semantic segmentation described below (see Figure 3).

### 5.1. Higher Order Orthogonal Iteration (HOOI) for Spectral Image Compression

Quoting Kolda, "The truncated higher order singular value decomposition (HOSVD) is not optimal in terms of giving the best fit as measured by the norm of the difference, but it is a good starting point for an iterative alternating least square algorithm" [17]. HOOI is an iterative algorithm to compute a rank-$(R_1, \ldots, R_N)$ TKD. Let $\mathcal{X} \in \mathbb{R}^{I_1 \times \cdots \times I_N}$ be an $N$-th order tensor and $R_1, \ldots, R_N$ be a set of integers satisfying $1 \leq R_n \leq I_n$, for $n = 1, \ldots, N$; the rank$-(R_1, \ldots, R_N)$ approximation problem is to find a set of $I_n \times R_n$ matrices $\mathbf{U}^{(n)}$ column-wise orthogonal and a $R_1 \times \cdots \times R_N$ core tensor $\mathcal{G}$ by computing

$$\min_{\mathcal{G}, \mathbf{U}^{(1)}, \ldots, \mathbf{U}^{(N)}} ||\mathcal{X} - \mathcal{G} \times_1 \mathbf{U}^{(1)} \cdots \times_N \mathbf{U}^{(N)}||^2, \tag{15}$$

and from matrices $\mathbf{U}^{(n)}$, where $\mathbf{U}^{(n)\mathrm{T}} \mathbf{U}^{(n)} = \mathbf{I}^{(n)}$, the core tensor $\mathcal{G}$ is found to satisfy (2) [34]. For a third-order tensor decomposition, we can rewrite (4) as

$$\hat{\mathcal{X}} = \mathcal{G} \times_1 \mathbf{U}^{(1)} \times_2 \mathbf{U}^{(2)} \times_3 \mathbf{U}^{(3)} \tag{16}$$

where $\hat{\mathcal{X}}$ denotes the reconstruction approximation of the input spectral image $\mathcal{X}$, $\mathcal{G}$ is the $J_1 \times J_2 \times J_3$ core tensor, and $\mathbf{U}^{(1)} \in \mathbb{R}^{I_1 \times J_1}$, $\mathbf{U}^{(2)} \in \mathbb{R}^{I_2 \times J_2}$ and $\mathbf{U}^{(3)} \in \mathbb{R}^{I_3 \times J_3}$ are the projection matrices. Algorithm 1 shows HOOI for a third order tensor decomposition, but the extension to higher order tensors is straightforward. Thus, with Algorithm 1 we compute the tensor $\mathcal{G}$ with rank-$(J_1, J_2, J_3)$ for each spectral image as third-order tensor.

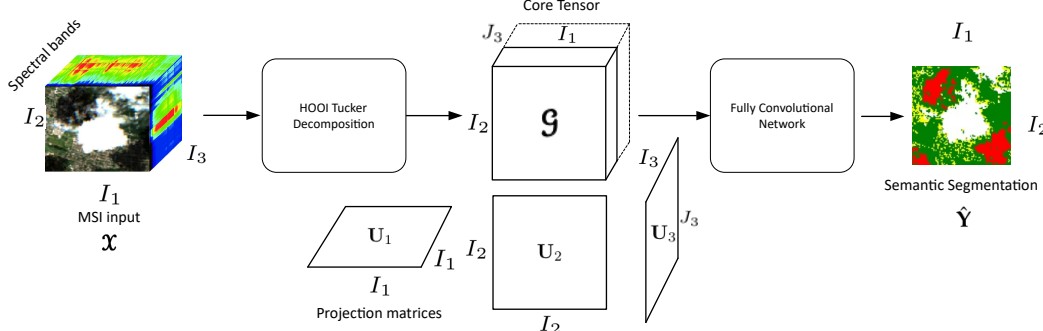

**Figure 3.** The big picture of the fast semantic segmentation framework proposed, with a fully convolutional network encoder-decoder architecture and a preprocessing HOOI tucker decomposition stage.

---

**Algorithm 1:** HOOI for MSI. ALS algorithm to compute the core tensor $\mathcal{G}$.

---

**Function** HOOI($\mathcal{X}$, $R_1$, $R_2$, $R_3$)**:**

> initialize $\mathbf{U}^{(n)} \in \mathbb{R}^{I_n \times R_n}$ for $n = 1, 2, 3$ using HOSVD;
> **repeat**
> > **for** $n = 1, 2, 3$ **do**
> > > $\mathcal{D} \leftarrow \mathcal{X} \times_1 \mathbf{U}^{(1)\text{T}} \times_2 \mathbf{U}^{(2)\text{T}} \times_3 \mathbf{U}^{(3)\text{T}}$
> > > $\mathbf{U}^{(n)} \leftarrow R_n$ leading left singular vectors of $\mathbf{D}_{(n)}$
> > **end**
> **until** *fit ceases to improve or maximum iterations exhausted;*
> $\mathcal{G} \leftarrow \mathcal{X} \times_1 \mathbf{U}^{(1)\text{T}} \times_2 \mathbf{U}^{(2)\text{T}} \times_3 \mathbf{U}^{(3)\text{T}}$

**Output:** $\mathcal{G}, \mathbf{U}^{(1)}, \mathbf{U}^{(2)}, \mathbf{U}^{(3)}$

---

*5.2. Fcn for Semantic Segmentation of Spectral Images*

We use a FCN model for semantic segmentation based on the proposed by Badrinarayanan et al. in [35] called Segnet. Each core tensor $\mathcal{G}$ obtained after decomposition, is the input to the SegNet for training and testing the network. Hence, the feature activation maps $\mathcal{M} \in \mathbb{R}^{I_1 \times I_2 \times F_2}$ for each hidden layer of the SegNet encoder-decoder FCN are computed by displacing the filters $\mathcal{W}$ through the whole input core tensor in strides $S = 1$. It is worth noting that kernel $\mathcal{W}$ is a four-order tensor $\mathcal{W} \in \mathbb{R}^{K_1 \times K_2 \times F_1 \times F_2}$, where $K_1$ and $K_2$ represent its spatial dimensions height and width; $F_1$ its depth, i.e., the spectral domain; and $F_2$ denotes the number of filters used to produce $F_2$ activation maps (Figure 2). We express this convolution operation as

$$\mathbf{M}^{(f_2)} = \sigma\left(\mathcal{W} \odot \mathcal{G}\right), \tag{17}$$

where $\mathbf{M}^{(f_2)}$ represents each activation map for $f_2 = 1, \ldots, F_2$, and each value $m_{i_1 i_2 f_2}$ is computed as in (12). $\sigma$ denotes the rectified linear unit (ReLU) [33] function; i.e., $\sigma(z) = \max\{0, z\}$. Symbol $\odot$ is used in this paper to represent the convolution; i.e., the whole operation applied in convolutional layers (see Figure 2). These activation maps are the input for the subsequent layer in the SegNet FCN.

The last layer is used the softmax activation function [33] to produce a distribution probability, and so, predict values relating each pixel to one of the $C$ classes of interest. Hence, for the last layer we rewrite (17) as

$$\hat{\mathbf{Y}} = \delta\left(\mathcal{W} \odot \mathcal{M}\right), \tag{18}$$

where $\hat{\mathbf{Y}}$ represents the output prediction, $\mathcal{M}$ is the feature activation maps at previous layer, $\delta$ the softmax activation function, and $\mathcal{W}$ the filter or kernel tensor with the adaptable synaptic weights.

The output of the FCN is a matrix $\hat{\mathbf{Y}}$ with the same spatial dimensions as the input, with a value of the most likely class for each pixels. Figure 4 shows the architecture of the SegNet model used in

this work. Experiments present the behavior of this FCN with and without data compression in the spectral domain.

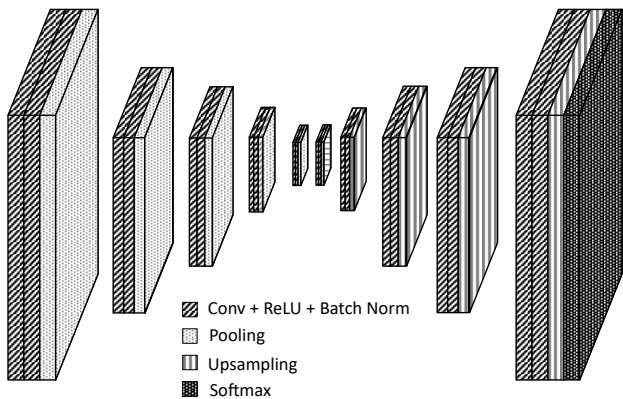

**Figure 4.** SegNet FCN. Encoder-decoder architecture with convolutional, pooling, and upsampling layers with their corresponding activation functions and batch normalization [33].

## 6. Experimental Results

### 6.1. Our Data

As case study, a CNNMSI dataset with 100 RS images was used for training and 10 for testing, all of them from central Europe with $128 \times 128$ pixels. These images are partitions of the original Sentinel-2 images without modification and all semi-manually labeled, and with abundant presence of the elements of interest. In Table 3 the 10 scenarios correspond to our 10 images for testing. We used only nine from the 13 available spectral bands from visible, NIR to SWIR wavelengths. Bands 2, 3, 4, and 8 have 10 m resolution, and bands 5, 6, 7, 11, and 12 have 20 m (oversampled to 10 m [18]). These bands provide decisive information for discrimination of different classes. Bands 1, 9, and 10 were dismissed because of their lower spatial resolution of 60 m. Band 8A, also with 20 m spatial resolution, was dismissed due to wavelength overlapping with band 8. It is worth mentioning that the framework proposed in this work can be applied to any kind of spectral image and multitemporal datasets [36].

#### 6.1.1. The Training Space

For training, the input data ws a tensor $\mathfrak{X} \in \mathbb{R}^{128 \times 128 \times 9 \times 100}$, where $128 \times 128$ is the spatial dimensions, 9 is the number of spectral bands, and 100 is the number of images used for training. Although the number of images seems low, taking into account that we work at pixel-domain, the real number of training points or vectors is high. Indeed, our FCN for semantic segmentation was trained with $128 \times 128 \times 100 = 1638400$ samples or vectors. To test whether the size of the data for training was sufficiently high, a smaller subtensor of $\mathfrak{X}$, $\mathfrak{X}_p \in \mathbb{R}^{128 \times 128 \times 9 \times 80}$, equivalent to 1310720 points or vectors, was used for a second training obtaining, for the same test set, an average PA of 91.48%; i.e., only 0.08% less than with 100 images, 91.56%. We also tested these results by a third training with an extended dataset of 120 images, $\mathfrak{X}_q \in \mathbb{R}^{128 \times 128 \times 9 \times 120}$ equivalent to 1966080 vectors, and we found only a slight variation of $+0.01\%$ in the PA (91.57%), while the execution time for the training increased significantly.

#### 6.1.2. The Labels

Our labels were acquired using the scene classification algorithm developed by the ESA [19], and subsequently modified, semi-manually, misclassified pixels.

### 6.1.3. The Testing Space

For testing, our input data were a $128 \times 128 \times 9 \times 10$ tensor; i.e., 10 different scenarios for pixel-wise classification, whose results are shown in Table 3. That is, the framework classifies $128 \times 128 \times 10 = 163,840$ pixels.

### 6.1.4. Downloading Data

Due to the big size of the data, format npy was used. Data are available in the link Dataset.

- The training dataset is in the file S2_TrainingData.npy.
- Labels of the training dataset are in the file S2_TrainingLabels.npy.
- A true color representation of the training dataset can be found in S2_Trainingtruecolor.npy.
- The testing dataset and the corresponding labels are in the file S2_TestData.npy.
- Labels of the test dataset are in the file S2_TestLabels.npy.
- Last, a true color representation of the test data can be found in S2_Testtruecolor.npy.

Code will be delivered by the corresponding author upon request for research purposes only.

### *6.2. Classes*

The CNNMSI dataset has been semi-manually labeled for supervised semantic segmentation of $C = 5$ classes; vegetation, water, cloud, cloud shadow, and soil. These classes were selected according to their impact in RS research areas such as agriculture, forest monitoring, population growth analysis, and disaster prevention. It is worth mentioning that the detection of clouds and cloud shadows is an important prerequisite for almost all RS applications.

### *6.3. Metrics*

### 6.3.1. Pixel Accuracy (PA)

We used the PA metric to compute a ratio between the amount of correctly classified pixels and the total number of pixels as

$$PA = \frac{\sum_{c=0}^{C} p_{cc}}{\sum_{c=0}^{C} \sum_{d=0}^{D} p_{cd}} \tag{19}$$

where we have a total of $C$ classes and $p_{ii}$ is the amount of pixels of class $c$ correctly assigned to class $c$ (true positives), and $p_{cd}$ is the amount of pixels of class $c$ inferred to belong to class $d$ (false positives). We can see in Table 3 the PA values for our proposed framework in comparison with other state-of-the-art methods. From Table 3, we can see that:

- Indexes NDI are important references for pixel-wise classification but they show one of the lowest PAs and the highest computational time.
- Classic PCA with five components shows the lowest PA, although the computational time is similar to HOOI-FCN with five tensor bands.
- Due to the poor results of NDI and classical PCA, FCN (with raw data and nine components) is a good reference in terms of performance and computational time, and HOOI-FCN with seven and five tensor bands achieves the highest PA and the lowest computational time.

The PA and the computational times for FCN and HOOI-FCN with different numbers of tensor bands are shown in Figure 5.

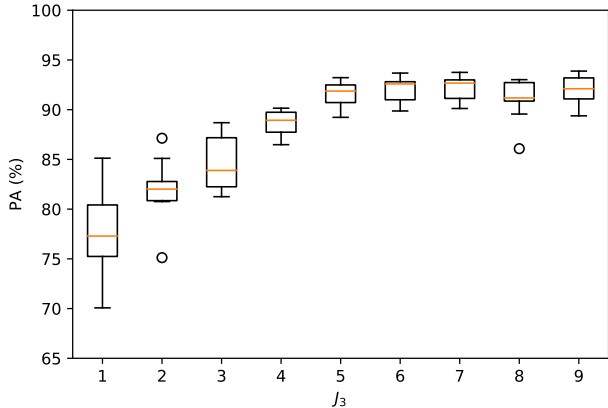

**Figure 5.** Box and whiskers plot of the pixel accuracy (PA) for the 10 testing scenarios shown in Table 3.

### 6.3.2. Relative Mean Square Error (rMSE)

In order to compute the reconstruction error of the tensor $\mathcal{X}$ for the implementation of HOOI, the rMSE was used:

$$rMSE\left(\hat{\mathcal{X}}\right) = \frac{1}{Q} \sum_{q=1}^{Q} \frac{\left\|\hat{\mathcal{X}}_q - \mathcal{X}_q\right\|_F^2}{\left\|\mathcal{X}_q\right\|_F^2}, \tag{20}$$

where $\mathcal{X}_q$ represents the $q$-th CNNMSI from our dataset with $Q$ MSIs and $\hat{\mathcal{X}}_q$ its corresponding reconstruction computed by (4).

Figure 6a shows the behavior of the reconstruction rMSE for our 100 training images for $J_3 = 1, \ldots, I_3$. With this metric we can quantify how good the decomposition represents the input data. The rMSE is also one of the decisive parameters to set the value of the $rank_3(\mathcal{X}) = J_3$. To preserve a high performance in the pixel-wise classification task, we set the threshold $\psi$ to a value for which the rMSE error is less than or equal to 0.05%, since deeper decomposition decrease the PA to less than 90%, as we can see in Figure 5. For a rank decomposition (128, 128, 5) our rMSE is 0.04%, which means that we reduce the dimensionality of our input data to almost half with a very low loss in performance. Besides, comparing this error with matrix based methods as PCA, we can see that our tensor-based decomposition produces lower rMSE for every value of $J_3$ except for the first one.

### 6.3.3. Orthogonality Degree of Factor Matrices and Tensor Bands

A way to analyze the algorithm HOOI efficiency is computing the orthogonality degree of the core tensor $\mathcal{G}$ and the projection matrices $\mathbf{U}^{(n)}$. As we mentioned in Section 3, we use the all-orthogonality property proposed in [32] and described in (7) and (8) to evaluate the orthogonality degree of our core tensors. Table 4 shows the results of the inner products between each tensor band with the others from one of our training images. We can see that these values are practically zero, which means that our bands are orthogonal. Furthermore, we can see in Figure 6b that (8) is fulfilled.

It is also important to know the orthogonality degree in our projection matrices. From Theorem 2 in [32] we start from the condition $\mathbf{U}^{(n)\mathrm{T}}\mathbf{U}^{(n)} = \mathbf{I}^{(n)}$; then, we create a vector $\hat{\mathbf{o}}$ where the components are the trace of each resulting matrix, i.e., $\mathrm{tr}(\mathbf{I}^{(n)})$, and compute the MSE with respect to a vector rank $\mathbf{o} = (J_1, J_2, J_3)$ as

$$MSE(\hat{\mathbf{o}}) = \sum_{q=1}^{3} \left\|o_q - \hat{o}_q\right\|_F^2. \tag{21}$$

Using this orthogonality analysis, we obtain MSE values very close to zero, e.g., in order of $10^{-20}$, which means that projection matrices present a high orthogonality degree.

**Table 3.** Quantitative results for 10 test MSIs running in a NVIDIA GeForce GTX 1050 Ti GPU, Intel core i7 processor, 8 Gb RAM, SSD 128 Gb, and HDD 1 Tb. Values in blue and red represent the highest PA and the lowest time, respectively.

| Scenarios | NDI | | FCN$_9$ | | PCA-FCN$_5$ | | HOOI-FCN$_7$ | | HOOI-FCN$_5$ | |
|---|---|---|---|---|---|---|---|---|---|---|
| | PA (%) | Time (s) | PA (%) | Time (s) | PA (%) | Time (s) | PA (%) | Time (s) | PA (%) | Time (s) |
| 1 | 88.20 | 363.03 | 91.05 | 101.21 | 85.12 | 9.85 | 91.12 | 37.84 | 90.63 | 9.13 |
| 2 | 84.75 | 412.89 | 92.21 | 87.54 | 84.60 | 9.83 | 90.12 | 36.54 | 89.23 | 9.06 |
| 3 | 92.34 | 307.56 | 93.67 | 93.45 | 88.32 | 10.00 | 93.75 | 36.02 | 93.22 | 9.03 |
| 4 | 90.08 | 382.31 | 91.72 | 98.92 | 86.08 | 9.73 | 92.85 | 36.79 | 92.18 | 8.93 |
| 5 | 87.14 | 400.12 | 89.91 | 103.57 | 86.36 | 9.12 | 92.13 | 35.88 | 91.84 | 9.67 |
| 6 | 89.75 | 312.15 | 90.95 | 95.21 | 87.65 | 10.15 | 92.95 | 37.23 | 92.71 | 10.09 |
| 7 | 85.73 | 373.84 | 89.92 | 107.13 | 88.47 | 9.63 | 93.06 | 35.56 | 92.59 | 9.55 |
| 8 | 91.49 | 308.00 | 90.17 | 95.45 | 85.78 | 9.76 | 90.23 | 36.34 | 90.12 | 9.14 |
| 9 | 89.38 | 397.92 | 90.74 | 80.33 | 87.91 | 10.26 | 92.50 | 37.09 | 92.18 | 10.11 |
| 10 | 90.01 | 352.66 | 88.52 | 112.85 | 84.32 | 9.88 | 91.17 | 35.53 | 90.97 | 9.85 |
| **Average** | **88.87** | **361.04** | **90.88** | **97.56** | **86.46** | **9.82** | **91.97** | **36.48** | **91.56** | **9.45** |

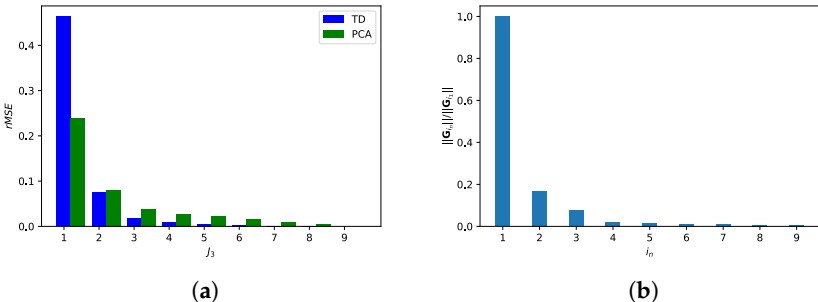

(a)        (b)

**Figure 6.** TD metrics (**a**) Reconstruction error computed by the relative mean square error (rMSE) for $J_3 = 1, ..., I_3$ and (**b**) norm of each subtensor $\mathcal{G}_{i_n}$, relative to the norm of the first tensor band $\mathcal{G}_{i_1}$.

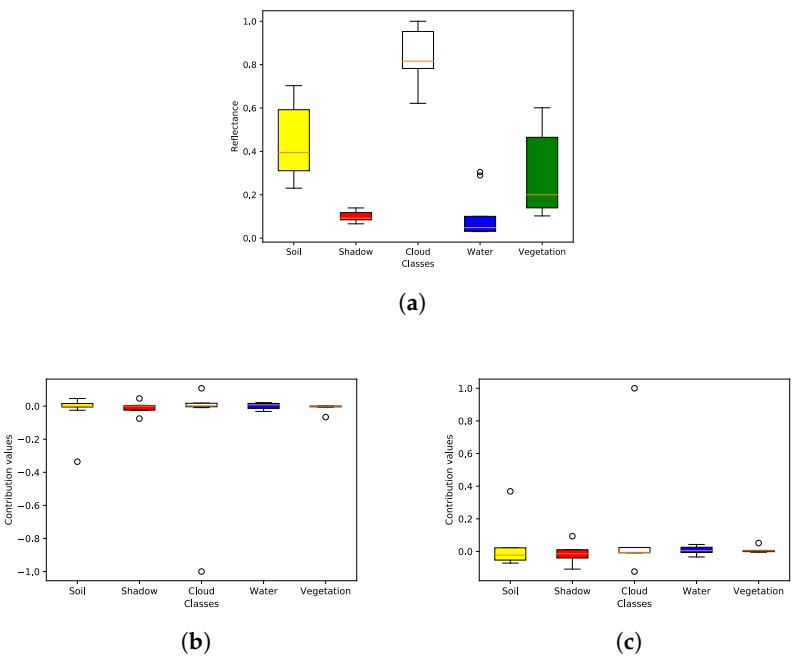

**Figure 7.** Box and whiskers plots of the behavior of five classes of interest: (**a**) in the original spectral domain, (**b**) the tensor band domain after decomposition for nine bands, and (**c**) the new tensor band domain for five bands.

### 6.4. Fcn Specifications

We used hyperparameter search [33] to set the learning rate to $1 \times 10^{-3}$. The model was run 100 epochs introducing 100 CNNMSI from our dataset. We used the Adam optimizer as our optimization algorithm. Xavier initialization was used for setting the initial values of the weights in the model. The Segnet FCN was used as the base model, since it achieves very high performance metrics in semantic segmentation [35].

### 6.5. Hardware/Software Specifications

Our framework was implemented using Python 3.7 with Tensorflow-GPU version 1.13. Experiments were run with a NVIDIA GeForce GTX 1050 Ti GPU. The processor used was an Intel core i7 with 8GB RAM, 128 GB SSD, and 1 TB HDD.

**Table 4.** Inner products of each tensor band with the others from one image of our dataset decomposed by HOOI.

| Tensor Band | 1 | 2 | 3 | 4 | 5 | 6 | 7 | 8 | 9 |
|---|---|---|---|---|---|---|---|---|---|
| 1 | - | $2.7 \times 10^{-4}$ | $8.0 \times 10^{-5}$ | $7.0 \times 10^{-5}$ | $4.1 \times 10^{-5}$ | $9.7 \times 10^{-6}$ | $2.0 \times 10^{-5}$ | $2.6 \times 10^{-5}$ | $8.6 \times 10^{-5}$ |
| 2 | - | - | $3.1 \times 10^{-7}$ | $8.5 \times 10^{-6}$ | $4.9 \times 10^{-6}$ | $3.2 \times 10^{-6}$ | $3.6 \times 10^{-6}$ | $6.0 \times 10^{-6}$ | $4.8 \times 10^{-6}$ |
| 3 | - | - | - | $8.4 \times 10^{-7}$ | $3.9 \times 10^{-7}$ | $4.4 \times 10^{-7}$ | $4.1 \times 10^{-7}$ | $1.8 \times 10^{-9}$ | $1.0 \times 10^{-6}$ |
| 4 | - | - | - | - | $5.0 \times 10^{-8}$ | $2.6 \times 10^{-7}$ | $1.2 \times 10^{-8}$ | $5.3 \times 10^{-8}$ | $1.2 \times 10^{-7}$ |
| 5 | - | - | - | - | - | $3.7 \times 10^{-9}$ | $8.3 \times 10^{-9}$ | $2.6 \times 10^{-8}$ | $8.9 \times 10^{-9}$ |
| 6 | - | - | - | - | - | - | $1.4 \times 10^{-8}$ | $7.2 \times 10^{-8}$ | $2.1 \times 10^{-7}$ |
| 7 | - | - | - | - | - | - | - | $1.2 \times 10^{-8}$ | $1.3 \times 10^{-9}$ |
| 8 | - | - | - | - | - | - | - | - | $1.6 \times 10^{-7}$ |

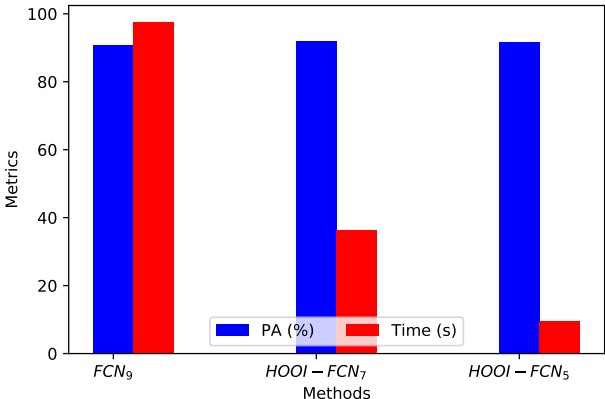

**Figure 8.** Comparison of the PA and the computational time of FCN with the proposed HOOI-FCN (seven and five bands) for semantic segmentation. See Table 3.

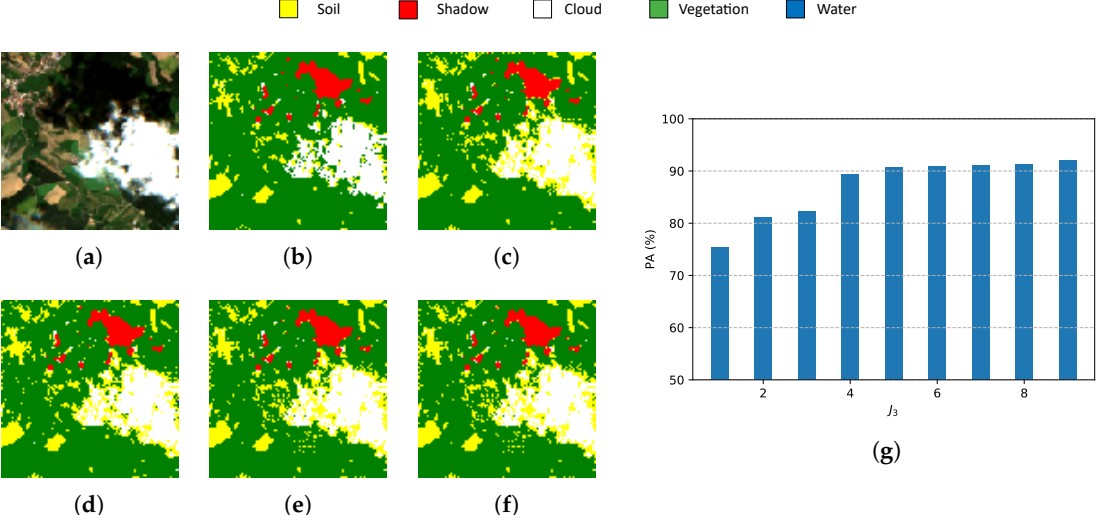

**Figure 9.** Qualitative results testing a scene of interest with abundant vegetation, and presence of shadows and clouds. (**a**) Original true color scenario of 128 × 128 pixels, in Central Europe: (**b**) five classes semi-manually labeled ground truth of the MSIs, (**c**) classification with an unsupervised normalized difference index (NDI) fusion algorithm, and (**d**) output prediction after 100 epochs in the FCN used for this work without data compression. (**e**) PCA-FCN framework output; (**f**) prediction of the whole framework HOOI-FCN proposed in this work; and (**g**) PA behavior of the HOOI-FCN versus number of tensor bands.

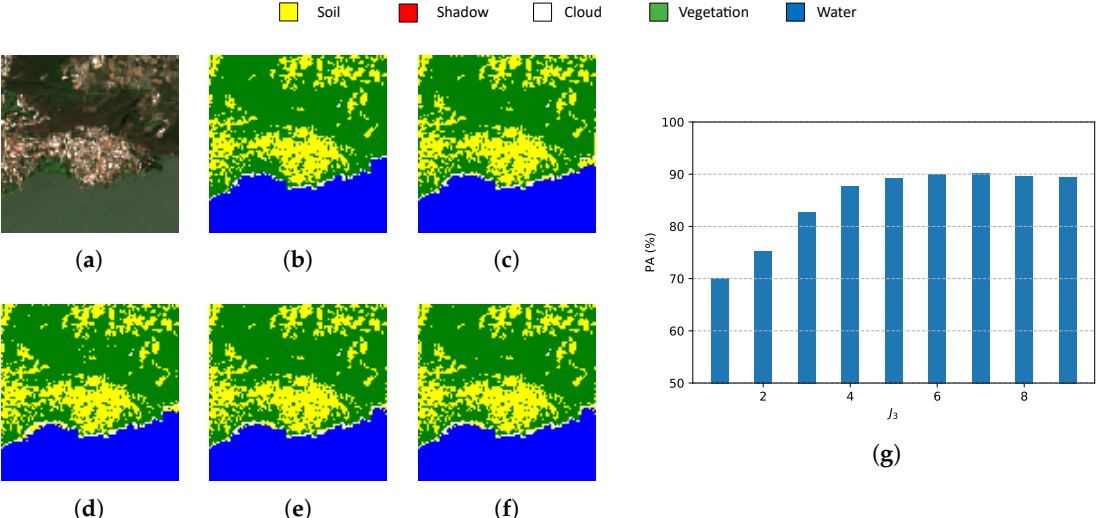

**Figure 10.** Qualitative results testing a scene of interest with abundant vegetation, and presence of shadows and clouds. (**a**) Original true color scenario of 128 × 128 pixels, in Central Europe: (**b**) five classes semi-manually labeled ground truth of the MSIs, (**c**) classification with an unsupervised normalized difference index (NDI) fusion algorithm, and (**d**) output prediction after 100 epochs in the FCN used for this work without data compression. (**e**) PCA-FCN framework output; (**f**) prediction of the whole framework HOOI-FCN proposed in this work; and (**g**) PA behavior of the HOOI-FCN versus number of tensor bands.

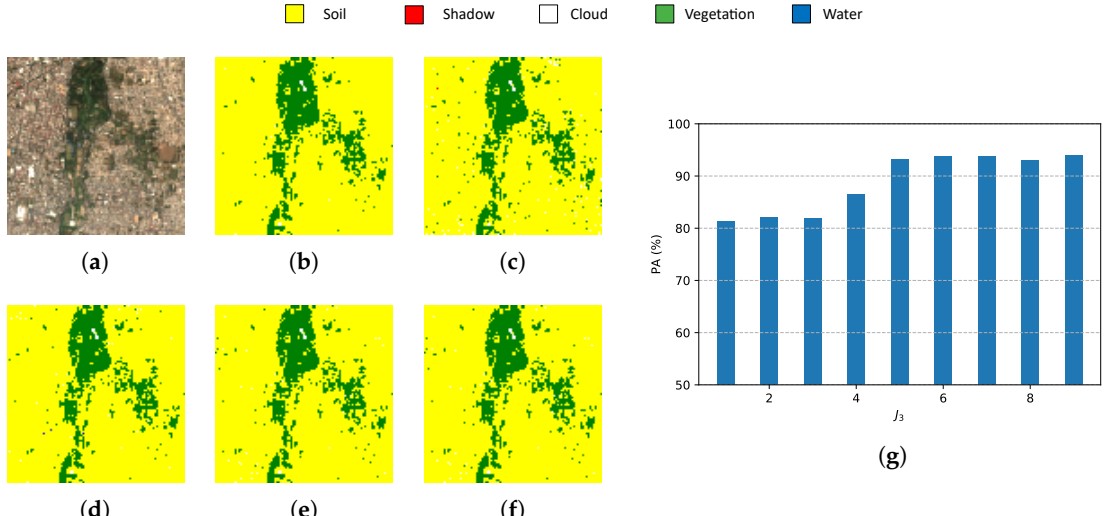

**Figure 11.** Qualitative results testing a scene of interest with abundant presence of soil. (**a**) Original true color scenario of 128 × 128 pixels, in Central Europe: (**b**) five classes semi-manually labeled ground truth of the MSIs, (**c**) classification with an unsupervised normalized difference index (NDI) fusion algorithm, and (**d**) output prediction after 100 epochs in the FCN used for this work without data compression. (**e**) PCA-FCN framework output; (**f**) prediction of the whole framework HOOI-FCN proposed in this work; and (**g**) PA behavior of the HOOI-FCN versus number of tensor bands.

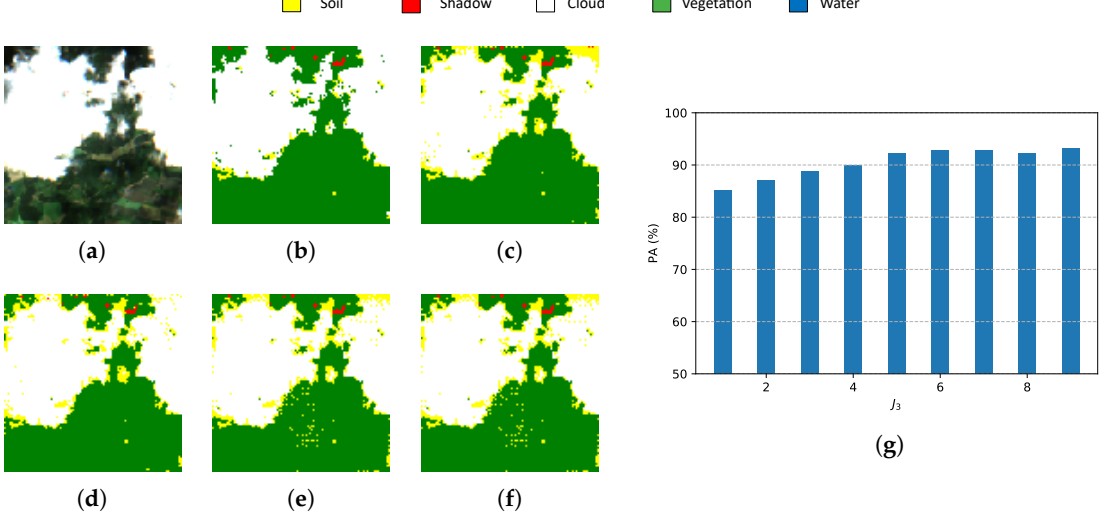

**Figure 12.** Qualitative results testing a scene of interest with abundant presence of clouds. (**a**) Original true color scenario of 128 × 128 pixels, in Central Europe: (**b**) five classes semi-manually labeled ground truth of the MSIs, (**c**) classification with an unsupervised normalized difference index (NDI) fusion algorithm, and (**d**) output prediction after 100 epochs in the FCN used for this work without data compression. (**e**) PCA-FCN framework output; (**f**) prediction of the whole framework HOOI-FCN proposed in this work; and (**g**) PA behavior of the HOOI-FCN versus number of tensor bands.

## 7. Discussion and Comparison with Other Methods

Original spectral bands (Figure 7a) were transformed or mapped into new tensor bands (Figure 7b,c) which preserved features of our classes of interest within the first tensor bands, avoiding the use of all the original spectral bands, thereby reducing computational load in further applications.

From Figure 7b,c, we can see that, for the classes of interest in this case study, the error margin selected $\psi$ is indeed a good parameter to restrict the rank in the third mode, since the spectral information for differentiation of these five classes is a greater proportion than the first elements of the

spectral domain. Nevertheless, if a smaller value for $J_3$ were used, there would be a trade off in the performance of the semantic segmentation.

Quantitative results in Figures 8–12 and Table 3 present a comparison of the processing time and PA from our proposed framework with a model without any preprocessing data decomposition algorithm and with a normalized differentiation index based method in different scenarios. The accuracy values obtained by the proposed HOOI-FCN framework are better in overall than those obtained by the other methods under same conditions and scenarios, but with a quite significant decrease of the processing time, in the order of 10 times. It is worth noting that our HOOI-FCN framework with seven and five tensor bands outperforms in PA to the same FCN with the original nine bands. This means that the decomposition produces better features for the classification ANN.

In the confusion matrix presented in Figure 13, we can see the accuracy of the framework proposed HOOI-FCN for each class and the overall accuracy. Rows correspond to the output class or prediction and the columns to the truth class. Diagonal cells show the correctly classified pixels. Off-diagonal cells show where the errors come from. The rightmost column shows the accuracy for each predicted class, while the bottom row shows the accuracy for each true class. It is important to note that vegetation and cloud classes are close to 95% accuracy, while for water and cloud shadows have less than 90% accuracy. The latter can be caused by the lack of samples with a greater contribution of these elements in the training dataset as well as the similarity of these elements to others in the scenes.

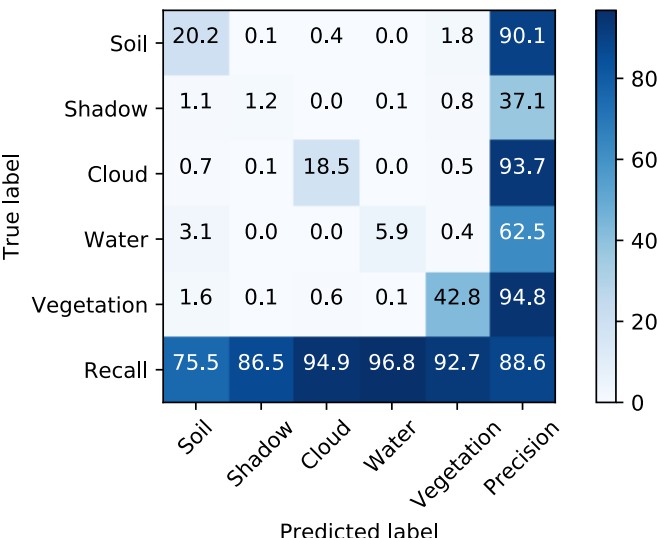

**Figure 13.** Confusion matrix of the proposed framework. The main diagonal indicates the pixel accuracy for each class in % for the ten selected scenarios.

## 8. Conclusions

Any RS-MSI or -HSI or third-order tensor image is mapped by the TKD to another tensor, called core tensor representative of the original, preserving its spatial structure, but with fewer tensor bands. In other words, a new subspace embedded in the original space was found and it was be used as the new input space for the task of pixel-level classification or semantic segmentation. Due to the success of DL for image processing, our approach employs an FCN network as the classifier, which delivers the corresponding prediction matrix of pixels classified element-wise.

The efficiency of the proposed higher order orthogonal iteration (HOOI)-FCN framework is measured by metrics such as pixel accuracy (PA) or recall as a function of the number of new tensor bands, which is defined by the reconstruction error computed by the rMSE. Another important parameter in the TKD is the orthogonality degree of each component, i.e., the core tensor and the factor matrices, computed by the inner products of each band with the others.

Our experimental results for a case study show that the proposed HOOI-FCN framework for CNNMSI semantic segmentation reduced the number of spectral bands from nine to seven or five tensor bands, for which PA values converge or are very close to the maximum.

State-of-the-art methods, such as normalized difference indexes, PCA with five principal components, and the same FCN network with nine original bands, with an average pixel accuracy 90% (computational time ∼90s), were outperformed by the HOOI-FCN framework, which achieved a higher average pixel accuracy of 91.97% (and computational time ∼36.5s), and average PA of 91.56% (computational time 9.5s) for seven and five new tensor bands respectively.

These results are very promising in RS, since the use of other algorithms for the calculation of core tensors and a deeper data analysis of weights and initialization of the convolutional neural network (CNN) can increase performance metrics of the segmentation for RS spectral data. Some limitations for a better validation of this approach are: denoising is not included; there is a need for new cases to enhance the input space; use of a greater number of classifiers is needed.

Finally, this research allows us to emphasize two main, relevant points. (1) RS images are characterized by a large number of bands, high correlation between neighbor bands, and high data redundancy; (2) besides, they are corrupted by several noises. Some issues related to our approach remain open.

*Open Issues*

- Compression affects not only the input data, but also the CNN network to reduce overall complexity and/or create new ANN architectures for specific RS-CNNMSI or HSI image applications.
- Instead of the HOOI algorithm, use greedy HOOI and other algorithms that determine the core tensor for a broad comparison.
- For classification purposes, use other machine learning algorithms, such as a SVM or random forest.
- Increase the input data with more scenarios and their corresponding ground truth to a deeper study of the behaviors of several classifiers, including those based on ANN, and the scope of the TD methods.
- Denoise the original input data for an improvement of the new subspace of reduced dimensionality.

**Author Contributions:** Conceptualization, S.S.; formal analysis, C.A.; investigation, J.L.; methodology, J.L., D.T., and S.S.; resources, C.A.; software, J.L.; supervision, D.T. and C.A.; validation, S.S. and C.A.; writing—original draft, J.L. and D.T. All authors have read and agreed to the published version of the manuscript.

**Funding:** This work was supported by the National Council of Science and Technology CONACYT of Mexico under grants 280975 and 253955.

**Acknowledgments:** We would like to thank the student E. Padilla and his advisor A. Méndez for their help in producing the results for the PCA decomposition and facilitate the comparative analysis between TD and PCA; and F. Hermosillo for useful observations about this work, all of them being from CINVESTAV, Guadalajara. We also would like to dedicate this work to the memory of Y. Shkvarko, who was an important mentor for the realization of this research.

**Conflicts of Interest:** The authors declare no conflict of interest.

## Abbreviations

The following abbreviations are used in this manuscript:

| | |
|---|---|
| **ANN** | artificial neural network |
| **CNN** | convolutional neural network |
| **CPD** | canonical polyadic decomposition |
| **ESA** | european space agency |
| **DL** | deep learning |
| **FCN** | fully convolutional network |
| **GPU** | graphics processing unit |

| HSI | hyperspectral image |
|---|---|
| HOOI | higher order orthogonal iteration |
| HOSVD | higher order singular value decomposition |
| MSE | mean square error |
| ML | machine learning |
| MSI | multispectral image |
| NIR | near-infrared |
| NTD | nonnegative Tucker decomposition |
| NDVI | normalized difference vegetation index |
| NDWI | normalized difference water index |
| PA | pixel accuracy |
| PCA | principal components analysis |
| ReLU | rectified linear unit |
| rMSE | relative mean square error |
| RS | remote sensing |
| SVD | singular value decomposition |
| SWIR | short wave infrared |
| SVM | support vector machine |
| T-MLRD | tensor-based multiscale low rank decomposition |
| TD | tensor decomposition |
| TDA | tensor discriminant analysis |
| TKD | tucker decomposition |

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
