# Peer review of "Spectral Imagery Tensor Decomposition for Semantic Segmentation of Remote Sensing Data through Fully Convolutional Networks"

_remotesensing, doi:10.3390/rs12030517_

Round 1

Reviewer 1 Report

Article is interesting, very well written and I have read it with interest. The subject is current and has useful value. I highly appreciate the substantive value of the article. Please note the following:
1. Please note that all drawings have references in the text. It seems that the authors e.g. do not refer to figure 4.
2. Please explain what means scenarios 1, 2 ... 10 in table 4.
3. Please consider to move aim of research, presented in abstract, to the main text.

Reviewer 2 Report

Minors:
There is lack of line numbering so writting review is difficult.

Subsection 1.1 - citation of authors names should be identical.

"An ANN is trained by using iterative gradient-based optimizer" - name and reference should be added. It is in 6.4.

Fig.5 - there is box and whiskers plot for avarage, worst, etc.
Line continuity on this plot suggest (false) continuous relation.

Similar case for Fig.7 - how to interpret values between band numbers ?

6.5 false values
https://en.wikipedia.org/wiki/Gigabit
'b' is for bits

TensorFlow version and GPU symbol required.

Majors:

1.
"Dataset and code are available in 'Dataset and Code.'" - this not true and it leads to serious problems:
- only three data files are aviable (npy)
- code is not available
- sharepoint.com is not typical platform for sharing data and code, there is github and some others for large datasets
(data are volatile on such platfom)

2.
Dataset - 100 images with 128x128 resolution is very, very small dataset for training.
It should be proved that such small dataset is acceptable.

3.
Table.4 How to interpret values - there are ten tests, somtimes are results like bolded (best), sometimes far from best?
This table is misleading - 2D plot is required for the comparison of time and quality

Reviewer 3 Report

The paper deals with a proposed Higher-Order Orthogonal Iteration combined with the Full Conventional Network (HOOI-FCN) for Multispectral Image segmentation.

The idea and the main contribution are clearly presented; The theoretical formulations are well detailed; and the results are also well illustrated.

The work presented in this paper is interesting for the readers and the researchers work in the field of application of Deep-Learning techniques in remote sensing.

Round 2

Reviewer 2 Report

Review 2

Observation /Comment #4 - any line type create continuous relation between values

Box-and-whiskers plot is desired

Review 2

Observation /Comment #5 - any line type create continuous relation between values

Author Response

The last suggestions of the Reviewer 2 were attended:

Observation /Comment #4 - any line type create continuous relation between
values Box-and-whiskers plot is desired.

Observation /Comment #5 - any line type create continuous relation between
values.

See now a representation of the Figures 5 and 7 (box and whisker plot),
and Figures 9.g, 10g, 11g ang 12g (discrete domain)

Thanks again to the anonymous reviewers.